# Animal Models of Hypertension (ISIAH Rats), Catatonia (GC Rats), and Audiogenic Epilepsy (PM Rats) Developed by Breeding

**DOI:** 10.3390/biomedicines11071814

**Published:** 2023-06-24

**Authors:** Marina A. Ryazanova, Vladislava S. Plekanchuk, Olga I. Prokudina, Yulia V. Makovka, Tatiana A. Alekhina, Olga E. Redina, Arcady L. Markel

**Affiliations:** 1Federal Research Center, Institute of Cytology and Genetics, Siberian Branch of Russian Academy of Sciences, Novosibirsk 630090, Russia; ocean-2006@yandex.ru (M.A.R.); lada9604@mail.ru (V.S.P.); petrenko@bionet.nsc.ru (O.I.P.); muv_97@mail.ru (Y.V.M.); alek@bionet.nsc.ru (T.A.A.); markel@bionet.nsc.ru (A.L.M.); 2Department of Natural Sciences, Novosibirsk State University, Novosibirsk 630090, Russia

**Keywords:** animal model, hypertension, ISIAH rat strain, audiogenic epilepsy, catatonia, stereotypy, genetic catatonia rat strain, pendulum-like movements rat strain

## Abstract

Research into genetic and physiological mechanisms of widespread disorders such as arterial hypertension as well as neuropsychiatric and other human diseases is urgently needed in academic and practical medicine and in the field of biology. Nevertheless, such studies have many limitations and pose difficulties that can be overcome by using animal models. To date, for the purposes of creating animal models of human pathologies, several approaches have been used: pharmacological/chemical intervention; surgical procedures; genetic technologies for creating transgenic animals, knockouts, or knockdowns; and breeding. Although some of these approaches are good for certain research aims, they have many drawbacks, the greatest being a strong perturbation (in a biological system) that, along with the expected effect, exerts side effects in the study. Therefore, for investigating the pathogenesis of a disease, models obtained using genetic selection for a target trait are of high value as this approach allows for the creation of a model with a “natural” manifestation of the pathology. In this review, three rat models are described: ISIAH rats (arterial hypertension), GC rats (catatonia), and PM rats (audiogenic epilepsy), which are developed by breeding in the Laboratory of Evolutionary Genetics at the Institute of Cytology and Genetics (the Siberian Branch of the Russian Academy of Sciences).

## 1. ISIAH (Inherited Stress-Induced Arterial Hypertension) Rats

Arterial hypertension is a widespread disorder that can lead to fatal complications; therefore, understanding the pathogenesis and prevention of hypertension is very important. Despite many years of research, hypertension remains a major medical problem, and the primary causes and mechanisms of essential hypertension are still unclear. This is because hypertension is a complex multifactorial disorder that has a polygenic basis and interacts with many environmental factors, including social and psychosocial stressors. Therefore, the creation of an animal model of stress-sensitive arterial hypertension is a useful achievement that may help to clarify the pathogenesis and pathophysiology of arterial hypertension.

### 1.1. A Short History of the ISIAH Rat Strain

The ISIAH rat strain was obtained by using genetic selection from an outbred normotensive Wistar rat colony. Systolic blood pressure (BP) is measured by using the tail-cuff method. The basal BP is determined when a rat is anesthetized for a short time (several minutes) with ether to exclude the influence of the BP measurement procedure on the basal BP level. The response of BP to stress has been evaluated in unanesthetized rats after 30 min of confinement in a wire-mesh cylindrical cell (restraint stress). The detailed history of the selection procedure and establishment of the ISIAH rat strain is presented in [1]; here, we only provide a brief description. The selection was started in 1972. The mean basal BP level in the original Wistar rat population was 118 mmHg (*n* = 283). In some rats, however, the stress-induced BP increased to 150 mmHg or even higher. This made it possible to begin genetic selection for the stress-induced hypertensive response in rats. As a result of crossings of closely related rats in several tens of generations, an inbred strain of rats with stress-sensitive arterial hypertension (named the ISIAH rat strain) was obtained. Of note, the selection of an enhanced BP response to stress also led to an increase in the basal BP. Long-term measurements of BP in ISIAH rats show that at the age of 3–4 months, the systolic BP is lower in females than in males by 10–15 mmHg [2]. Recent studies have mainly been conducted on males. Currently, the mean basal BP in the male population of this strain is 170–180 mmHg, and in ISIAH male rats exposed to short-term restraint stress, the BP reaches 190–200 mmHg. Thus, the hypertensive status of ISIAH rats can be regarded as persistent arterial hypertension with significant aggravation occurring in stressful environments [2,3]. In addition to the above characteristics of elevated BP at rest and its sharp increase under conditions of short-term restraint stress, ISIAH rats exhibit many features that are characteristic of human hypertension. These include both neuroendocrine aberrations that are associated with an increase in the reactivity of the sympathoadrenal and hypothalamic–pituitary–adrenal systems as well as a number of morphophysiological indicators of a hypertensive state [2,3,4]. A description of the main strain-specific traits of ISIAH rats is given in Table 1. The major findings of recent years are described in more detail in the text below.

### 1.2. Characteristics of the Main Neuroendocrine Pathways

It is known that the neuroendocrine system plays a central role both in the regulation of stress and in the pathogenesis of arterial hypertension. The stress response is implemented via two main neuroendocrine pathways: the sympathoadrenal and hypothalamic–pituitary–adrenocortical pathways. The regulation of BP levels is closely related to both of these systems and to the functioning of the renin–angiotensin–aldosterone system (RAAS), which in turn is functionally linked to neuroendocrine regulation.

Signs of elevated reactivity in the hypothalamic–pituitary–adrenal system of ISIAH rats are already observed at the age of 3 weeks (prehypertensive period). A comparative study on the morphology of the adenohypophysis in ISIAH rats and normotensive control rats (WAG strain) revealed features of the ultrastructural organization of cells indicating their functional activation, which may be associated with natural stress during the transition to self-feeding [5].

At the age of 2 months, concentrations of excitatory (glutamine and glutamate) and inhibitory (GABA and glycine) neurotransmitters in the cerebral cortex and hypothalamus of ISIAH rats and normotensive Wistar rats were studied using nuclear magnetic resonance spectroscopy. The results of the analysis suggested a reduced excitability of the cerebral cortex and enhanced excitability of the hypothalamus in ISIAH rats. A positive correlation was found between the levels of excitatory neurotransmitters and the mean arterial BP, which is in agreement with the existing theories about the activation of the hypothalamic centers in arterial hypertension [52].

In adult rats under restraint stress, a significant increase in the transcription of genes that encode the central hormones of the pituitary–adrenocortical system has been shown for CRH in the hypothalamus and POMC in the pituitary gland. Additionally, under different types of stress, ISIAH rats manifested a significantly greater increase in the secretion of ACTH by the pituitary gland and corticosterone by the adrenal cortex as compared with control (WAG) rats [4]. In ISIAH rats, compared with WAG rats, an elevated concentration of aldosterone in blood plasma was detected, as were higher rates of the secretion of corticosterone, 11-dehydrocorticosterone, and deoxycorticosterone, which was measured in the blood flowing from the adrenal vein after its cannulation. The decrease in the 11-dehydrocorticosterone/corticosterone ratio observed in ISIAH rats indicates a reduced functional activity of type 2 11-β-hydroxysteroid dehydrogenase (11-β-HSD), which converts corticosterone into its inactive form, cortisone. The response of both aldosterone and corticosterone to exogenous ACTH administered to rats with a dexamethasone blockade of endogenous ACTH was significantly higher in hypertensive ISIAH rats than in normotensive rats [11]. The most important stimulator of aldosterone secretion, angiotensin II, is upregulated by RAAS activation.

The results of several studies indicate the presence of increased basal activity in the cerebral RAAS of ISIAH rats [17,53]. This observation is confirmed by the finding that the blockade of the brain RAAS lowers BP in ISIAH rats [18]. In contrast, in the kidneys of adult ISIAH rats, the RAAS is inhibited [9,19]. In blood plasma, concentrations of renin and angiotensin-converting enzyme (ACE) in ISIAH rats are unchanged. At the same time, a significant increase in the concentrations of angiotensin II and aldosterone in the blood serum has been noted [53]. Taking into account the above observations and the presence of increased secretory activity in the adrenal cortex of ISIAH rats, we can say that, when at rest, ISIAH rats are nevertheless characterized by elevated functional activity of the hypothalamic–pituitary–adrenal and sympathoadrenal systems as well as some specific features of the functioning of several other hormonal systems that are associated with the manifestation of the hypertensive status in ISIAH rats. Another important characteristic of ISIAH rats is their enhanced responsiveness to stressors [4,11].

To identify the molecular genetic mechanisms that determine the distinctive traits of the manifestation of the hypertensive status in ISIAH rats, comparative analyses of transcriptomes from the brain stem [40], hypothalamus [36], adrenal glands [38], and renal cortex and medulla [37,39] have been carried out. The functional annotation of genes showing inter-strain differences in transcription levels between hypertensive ISIAH and control WAG rats has revealed that many of these genes are associated with a stress response. These results confirm that the basal state of functional tension (stress) in the key physiological mechanisms that form the hypertensive phenotype of ISIAH rats is genetically determined.

### 1.3. Genetic Mapping of Hypothalamic Norepinephrine Concentration in ISIAH Rats and Its Relations with Other Traits

Previously, we performed a quantitative trait locus (QTL) analysis to identify genetic loci that were associated with the key traits that determine the manifestation of hypertensive status in ISIAH rats. The following traits were analyzed: BP at rest and during short-term (30 min) restraint stress; the increase in BP during stress; body weight; absolute and relative weights of target organs (heart, kidneys, and adrenal glands); plasma corticosterone concentration at rest and under stress; the elevation of plasma corticosterone concentration under stress; and the behavior of ISIAH rats in the open field test [14,31,33]. Recently, the hypothalamic norepinephrine concentration was mapped in adult ISIAH rats to determine the genetic loci that were associated with the increase in the concentration of norepinephrine in the hypothalamus; loci that were shared with other characteristic features of the hypertensive state (listed above) were identified as well [35]. The locus that was most statistically significantly associated with the concentration of norepinephrine in the hypothalamus was found on chromosome 18. This QTL proved to be associated with both an increase in the concentration of norepinephrine in the hypothalamus and a higher heart weight in ISIAH rats. Accordingly, this locus may contain genes that are involved in enhanced sympathetic myocardial stimulation in ISIAH rats. Nevertheless, this QTL was found to not be associated with control over BP. The locus (on chromosome 1) associated with both arterial BP and cardiac hypertrophy in ISIAH rats has been previously described by our group [33]. Thus, the development of heart hypertrophy in ISIAH rats is governed by different genetic loci, one of which (on chromosome 18) correlates with the concentration of norepinephrine in the hypothalamus; the other locus (on chromosome 1) is associated with high BP [35].

The locus on chromosome 18 that is associated with the concentration of norepinephrine in the hypothalamus is quite long. In its proximal part, the QTL overlaps with the loci associated with several traits of rat behavior in the open field test (locomotor activity in the first minute of the first test trial, locomotor activity at the periphery of the open field area, and rearing at the periphery of the open field area), and in the central part of the chromosome, it overlaps with a QTL for the latency period [14]. The open field test allows researchers to evaluate basic psychophysiological characteristics, such as the severity of fear and anxiety reactions, locomotor activity, and levels of exploratory and displacement activities [54]. The existence of a relationship between the level of norepinephrine in the hypothalamus and the locomotor activity of animals has been demonstrated in various experimental models [55,56,57], but genetic control of these relations has not been studied thus far. The mapping of norepinephrine concentration in the hypothalamus, which was carried out for the first time by our group, turns over a new leaf in the research on these relationships, as our results suggest that genes located in the QTL in the proximal part of chromosome 18 in rats can play a key role in these processes.

### 1.4. Similarities and Differences in the Genetic Background between ISIAH Rats and Other Hypertensive Rat Strains

During the genetic mapping of traits (QTL analysis), all of the above-mentioned traits of ISIAH rats were only partially mapped to the same genetic loci, just as in other hypertensive strains. Many trait-associated loci were found to be specific for ISIAH rats and were identified for the first time, implying the existence of differences in the genetic control of the analyzed traits between ISIAH rats and other hypertensive strains. Therefore, the results of the QTL analysis showed that there are both similarities and differences in the genetic background between ISIAH rats and other hypertensive rat strains.

A study on the genetic similarity of ISIAH rats with other known rat strains has also been conducted using the SNPs that were identified during a transcriptomic analysis of ISIAH rats. A comparison was performed using the genome sequences of 42 strains and sub-strains of rats, 11 of which emulate spontaneous or induced types of hypertension [58]. In that paper, 1849 SNPs were identified that are in the homozygous state in ISIAH rats and are absent in any of the 42 strains and sub-strains of the other rats, strongly indicating the presence of specific genetic determinants in the transcriptome of ISIAH rats. Nonetheless, the most interesting discovery is a set of 158 polymorphisms that are only present in hypertensive rat strains (in ISIAH rats and in one or more of 11 other hypertensive strains and sub-strains: FHH/EurMcwi, LH/MavRrrc, MHS/Gib, SBH/Ygl, SHR/OlaIpcv, SHRSP/Gla, SHR/NCrlPrin, SHR/NHsd, SHR/OlaIpcvPrin, SS/Jr, and SS/JrHsdMcwi) but are absent in the other analyzed rat strains (non-hypertensive ones) [21]. An aspect of particular interest is that the maximum frequency of the same SNPs in various hypertensive strains and sub-strains is 0.58 (i.e., only in 7 of 12 hypertensive strains/sub-strains) [22]. This result is consistent with the evidence that hypertension is an extremely genetically heterogeneous disorder, and this conclusion may be true for humans.

The estimation of the distances (by multivariate scaling) between the genotypes of hypertensive ISIAH/Icgn rats and the 11 hypertensive strains and sub-strains of the rats listed above has uncovered significant differences in the ISIAH genotype from the genotypes of all analyzed strains [21]. On the other hand, the genotype of ISIAH rats turned out to be quite similar to that of the OXYS [59] rat strain, which was also selected at the Institute of Cytology and Genetics of the Siberian Branch of the Russian Academy of Sciences (ICG SB RAS) from the same outbred stock of Wistar rats as the ISIAH rats. The selection of OXYS rats was based on a trait that was not related to BP; nevertheless, OXYS rats have moderately elevated BP [60]. The results described above suggest that in the human population, groups that have historically lived close to each other may have more similar types of arterial hypertension than populations that have been historically spatially separated.

The findings reviewed above allow us to conclude that the ISIAH rat strain represents an original model in which both the development of hypertension and the genetically determined enhanced responsiveness to stressors are determined by a specific genetic background. It follows from the foregoing brief description of the ISIAH rat strain that its stress-sensitive type of hypertension is one of the adequate models of arterial hypertension that develops in humans under the conditions of urbanization and increased social stress. The ISIAH rat strain is a natural and internationally recognized addition to the plethora of experimental models that are currently being investigated regarding genetic predisposition to hypertension in humans [61,62]. The evidence base accumulated to date on the neurophysiological and molecular genetic pathogenesis of stress-sensitive hypertension in ISIAH rats allow investigators to proceed to the identification of potential pharmacological targets in this form of hypertension [1,44]. Lately, transcriptomic data from ISIAH rats, along with sequencing data that are available from other models of hypertension, have been used to identify common genetic determinants of the manifestation of various types of hypertension [63] and other age-related diseases [64].

## 2. GC (“Genetic Catatonia”) Rats

The mental health of a population is the most important medical, biological, and social issue available; the problem affects 792 million people around the world [65]. Investigations into the pathogenesis of neuropsychiatric diseases in humans have a number of limitations; therefore, to solve emerging problems, it is advisable to use experimental animal models [66]. Creating an adequate experimental model for neuropsychiatric pathologies is urgently needed, but this task is difficult due to their multifactorial nature [67,68].

Catatonia (from the Greek katàtonos: strained, tense) is a neuropsychiatric syndrome characterized by movement disorders, which are manifested both as freezing and hyperexcitation (psychomotor agitation) [69,70]. Catatonia was considered a subtype of schizophrenia until a large amount of evidence accumulated indicating that catatonic reactions occur in patients with various mental disorders. This realization has led to the recognition of catatonia as an independent (nonspecific) syndrome [71,72]. Currently, catatonia occurs in catatonic schizophrenia [73,74], bipolar disorder [75], depression [76], autism [77,78,79], and anti-NMDA receptor encephalitis [80], as well as due to adverse effects or an overdose of certain drugs. It is known that the prevalence of catatonic manifestations in these psychopathologies is quite high and reaches 7–31% (for a review, see [81]). Despite the high prevalence of catatonia, treatments are still nonspecific and are not based on evidence-based criteria [82]; this state of affairs once again highlights the importance of studying the mechanisms of catatonic disorders.

### 2.1. A Short History of the GC Rat Strain

The reactions consisting of freezing or excitation in animals can be attributed to normal adaptive reactions. Nonetheless, an excessively pronounced predisposition to these reactions, i.e., an extremely low genetically determined threshold, can lead to pathological conditions. Considering this, a model of catatonia that was named as the GC rat strain was created by professor V.G. Kolpakov via a selection approach [69]. The selection for a predisposition to catatonic reactions began in 1976 at the ICG SB RAS. The breeding program started with the mating of outbred Wistar rats, some of which were prone to a spontaneous “hanging” on the cage ceiling in a vertical catatonic posture. Unfortunately, the selection for this trait was not successful, but it was observed that a different type of catatonic posture could be induced in some “hanging” animals by gently lifting them with a stick by their front paws into a corner of the cage (Figure 1). These rats maintained their posture when the stick was removed, demonstrating a cataleptic state (catalepsy is an immobile condition with specific muscle tone in which an animal (or person) fails to change the imposed postures, and it is one of the main symptoms of catatonia [83]). Further selection was carried out according to the following criteria: The rats were tested five times by lifting their front paws with a test stick. A rat was considered to be cataleptic if it kept the given posture for at least 10 s in three out of five trials [84].

Although the selection was performed to enhance cataleptic freezing, individuals appeared in the population of the selected rats in which hyperkinetic reactions predominated, i.e., an increased defensive reaction, nondirectional locomotor agitation, and vocalization [85,86]. Moreover, the same rat could react with either freezing or hyperexcitation responses during consecutive tests. This observation confirms the validity of GC rats as an adequate model of catatonia because the same symptoms are observed in patients with catatonic syndrome [69].

### 2.2. Features of the GC Rat Strain

In clinical practice, various catatonia rating scales are used to diagnose catatonia and quantify its severity [87], with the Bush–Francis Catatonia Rating Scale being the most accurate and most popular [88,89]. According to this scale, the following diagnostic criteria are assessed in GC rats: (1) hyperexcitation (extreme nondirectional locomotor activity), (2) stupor (immobility, lack of a response to stimuli), and (3) frozen posture/catalepsy. Some GC rats also exhibit waxy flexibility, negativism, and rigidity.

Cataleptic freezing occurs in both GC males and females [84]. Rats with a catatonic type of reaction are characterized by behavioral aberrations in various tests. In particular, strong fear and anxiety in an aversive situation as well as impaired development of instrumental behavior during food reinforcement are observed in GC rats [85,86,90]. Anomalies have also been revealed in the social behavior of GC male rats, particularly in the form of a decrease in social interactions in the three-chamber test. On the other hand, in relation to an intruding unfamiliar male, GC rats show significantly longer social exploration in the home cage [91]. This discrepancy in social activity in the above two tests may be explained by differences in the environmental conditions that affect the emotional state and motivation. In the Barnes maze test, GC rats perform much worse in the probe trial than control rats do, possibly indicating the presence of memory impairment and cognitive disturbances in GC rats [91]. Most of our research has been conducted on male GC rats. GC female rats show altered maternal behavior (spending more time with pups inside the nest) and altered daily activities in lactating females compared with Wistar (control) rats [92]. We can hypothesize that inter-strain differences in maternal behavior may be related to greater anxiety in GC rats. The elevated activity of lactating GC females at night is similar to that of rats with increased anxiety and depression-like behavior [93,94].

The excessive pathological reaction of GC rats, even to weak stimuli, may be attributed to a deficiency in the filtration of sensorimotor information in the central nervous system [86]. This notion is evidenced by a decrease in prepulse inhibition and enhanced startle reflex [86,95,96]. The deficit of prepulse inhibition is considered an endophenotype of neuropsychiatric diseases and is widely used in the characterization of new experimental animal models [97]. Plasma corticosterone levels are elevated in GC rats but can be reduced by antidepressants [98].

In our work, some steps have been taken to find pharmacological ways to correct the phenotypic abnormalities (associated with the manifestation of catatonia) acquired by GC rats during the original selection process. The oral administration of different polymorphs of glycine has a beneficial effect on the behavior of GC rats. Both α- and γ-polymorphs of glycine increase the exploratory activity in the open field test, but only the γ-form of glycine has been reported to have a beneficial impact on catalepsy and exploratory activity in the light–dark box test. In addition, this compound alleviates anxiety in the elevated plus maze test [99]. Treatment with D-serine has been shown to increase anxiety and reduce the locomotor activity of GC rats in the elevated plus maze test in contrast to a Wistar (control) rat group [100]. Moreover, a positive effect of imipramine administration has been demonstrated [98,101]. Nonetheless, the effects of the main drugs that are clinically used to treat catatonia, benzodiazepines, have yet to be elucidated.

Compared with the original Wistar population, GC rats manifest deviations in the size of brain structures; the area of the striatum in the right hemisphere is smaller, while the area of the cortex is larger. Furthermore, a more than twofold decrease in the area of anterior horns of lateral ventricles has been registered in GC rats [102].

The search for molecular markers of catatonia in GC rats has revealed a decrease in α1A adrenoreceptor mRNA expression in the medulla oblongata and midbrain and α2A adrenoreceptor mRNA overexpression in the frontal cortex, implying an alteration of the adrenoreceptor component of the noradrenergic system of the brain [86,96].

A brief description of all of the above traits that are associated with the manifestation of catatonia in GC rats is presented in Table 2.

### 2.3. Other Animal Models of Catatonia

Animal catatonic reactions are not only observed in rats but also among many vertebrates, and they are considered to be a type of passive defensive behavior. An animal in a state of catalepsy is able to maintain an uncomfortable position for a long time; depending on the species and situation, this can range from several seconds to many hours [103]. In this paper, only rodent models of catatonia will be considered.

The most widely studied class of models are models of drug-induced catatonia based on behavioral effects of antipsychotic drugs such as haloperidol [104,105,106,107]. These models have emerged because of the known risk of catatonia in patients taking first-generation antipsychotics [108]. The use of dopamine (D2) and α-adrenergic receptor antagonists such as haloperidol limits the investigation into the etiopathogenesis of catatonia to monoamines. Nevertheless, data have recently been accumulating on the participation of other neurotransmitter systems (including glutamatergic) in catatonia in anti-NMDA receptor encephalitis [80]. Antibodies to the NR1 subunit of NMDA glutamate receptors play a leading role in the pathogenesis of this disorder [109]. The clinical picture in most cases is characterized by psychotic symptoms, often with such phenomena as psychomotor hyperexcitation and/or stupor. In rodents, drugs that antagonize NMDAR function induce a cataleptic freeze and stereotypical behaviors [110]. Furthermore, catatonia can be induced by the administration of other substances with different mechanisms of action: arecoline [111], histidine [112], zolpidem [113], or benzodiazepines [114], as well as by benzodiazepine withdrawal [115], high concentrations of lipopolysaccharides [116], and other factors. Such a variety in the substances that cause catatonic reactions underscores the complexity of the etiopathogenesis of this syndrome. Several theories have been proposed based on the available evidence, but the pathophysiology of catatonia is still unclear [81,117,118].

In contrast to chemically induced catalepsy, which can be reproduced in almost any mouse or rat, nonpharmacological catatonia is a rare phenomenon. Catatonic freezing in animals can be caused by various mechanical stimuli, for example by pinching the neck (“pinch-induced” catalepsy). Pinch-induced catalepsy (demonstrated in rats and mice) is regarded as a change in muscle tone and is related to the nonresponsiveness to external stimuli [119,120,121]. An example of this type of catalepsy model is the ASC strain of mice, which is characterized by a high predisposition to pinch-induced catalepsy [122] combined with a set of depressive-like behavioral and physiological features [123]. Experimental catatonic freezing in animals can also be induced by other mechanical modalities such as centrifugation [124] or exposure to flickering light (photogenic catalepsy) [125]. There have been reports that Wistar Kyoto (WKY) rats exposed to acute 1 h restraint stress can show greater freezing in behavior tests [126]. Krushinsky–Molodkina (KM) rats have well-pronounced postictal catalepsy [127], as do rats with pendulum-like movements (PM strain) [128].

Unlike the models of catalepsy described above, GC rats tend to respond with cataleptic freezing in tests involving a weak stimulus. Cataleptic freezing of GC rats occurs without a painful stimulus (in contrast to pinch-induced catalepsy) and without a prior epileptic seizure, in contrast to Krushinsky–Molodkina and PM strains. Aside from catatonic freezing, in behavioral tests, GC rats can respond with catatonic arousal, which makes them the model that most adequately reflects the nature of catatonia in patients.

## 3. PM (“Pendulum-like Movements”) Rats

Epilepsy is a neurological disorder that is characterized by spontaneous, recurrent seizures. It is the third most common chronic brain disease.

Epilepsy is accompanied by depression, anxiety, and substantially higher morbidity and mortality [129,130]. Although the pathogenesis of epilepsy has been intensively studied for a long time, quite a high percentage of cases are not amenable to pharmacotherapy. The diversity of pathogenetic mechanisms of epilepsy requires designing new experimental models. There is a range of conditions under the umbrella term of epilepsy, where each condition has distinct acquired, genetic, and epigenetic etiopathogeneses and various distinct behavioral traits, electrographic signatures, and pharmacological profiles. In this regard, the modeling of epilepsy in animals is a complex task that requires an integrated approach.

Animal models of epilepsy involve either an induced or hereditary predisposition to different types of seizure [131]. For example, audiogenic seizures are a known phenomenon. The kind of model based on inherited predisposition to epilepsy (the PM rat strain) has been developed at the ICG SB RAS (Russia) and has a propensity for audiogenic epilepsy.

### 3.1. A Short History of PM Rats

In 1977, Kolpakov et al. described specific catatonic forms of behavior that occur in response to a mild emotional stressor [132,133] in albino Norway rats, and the behavior involved a stereotyped hyperkinesis by way of rhythmic side-to-side swings of the head and torso (Figure 2). The selection of rats from a Wistar population for well-pronounced pendulum-like movements as a putative hyperkinetic pole of catatonia began in 1987. The proportion of rats with pendulum-like movements became significantly higher than that in the Wistar control stock after the S2 generation of selection. The selection plateau occurred in the S5 generation of selection, when the manifestation of pendulum-like movements was achieved in 100% of the rats [84]. Later, however, it has been noticed that rats of the PM strain (an abbreviation for “pendulum-like movements”) [133] also have a predisposition to seizures caused by audiogenic stimuli; thus, PM rats demonstrate a shift from a catatonic to an epileptiform type of responses. This observation is in good agreement with the fact that, in some cases, human epilepsy is accompanied by stereotyped behavior [134].

### 3.2. Traits of the PM Rat Strain

The first manifestations of this hyperkinesis appear at the age of 3 weeks, reaching their peak at about 2 months. This pathology does not depend on sex and manifests itself equally in males and females [84].

Aside from pendulum stereotypy and audiogenic seizures, PM rats demonstrate certain specific behavioral traits in various tests. For instance, in the open field test, it has been shown that the dynamics of the locomotor activity of PM rats differs from those of Wistar rats; having a greater number of crossed squares in the first minute, PM rats show diminished locomotor activity during minutes 2–6 [135]. Large differences in locomotor activity in the first minute compared with subsequent minutes indicate increased emotional excitability. After exposure to an audiogenic stimulus, higher excitability is observed, which is manifested as erratic jumping and paroxysmal running followed by prolonged postictal catalepsy (see Table 3) [136,137].

PM rats are characterized not only by postictal catalepsy, which also occurs in other models of audiogenic epilepsy [138], but also by more pronounced pinch-induced catalepsy in pups compared with controls [128]. Such stupor is regarded as a manifestation of a catatonic reaction.

PM rats exhibit heightened offensive behavior in the resident–intruder test [91] and high aggressiveness in the glove test [137], which may confirm the likely relation between seizure predisposition and aggressiveness. People with epilepsy also commonly have symptoms of neurological or psychiatric illness, such as cognitive impairment, depression, anxiety, attention deficits, and aggressiveness [139,140,141,142].

The high emotional excitability of PM rats contributes to a decrease in attention to environmental stimuli, thereby complicating spatial orientation. For example, in the Morris water maze test, PM rats demonstrate longer platform search time and a reduction in the proportion of successful attempts to find the platform [143]. Moreover, it has been reported that PM rats do not employ a spatial strategy in the Barnes maze, possibly also indicating an impairment in their learning and spatial memory [91].

For the early diagnosis of a disease and its timely treatment, it is important to identify prodromal signs, i.e., symptoms that emerge before the onset of the disease. Such signs have been detected in PM rats during the early neonatal period, including delayed development of locomotor responses, increased immobility, a longer eyes-closed period, a shift in circular movements, a lag of body weight gain, and a greater manifestation of excitable responses, such as vocalizations and paroxysms [128].

In PM rats, a lower concentration of taurine in the hippocampus has been documented, which is related to a predisposition to convulsive conditions [102]. In addition, taurine, which is used in the treatment of epilepsy [144], alleviates audiogenic seizures in adult PM rats [145]. To date, some data have been obtained on the changes in monoamines’ levels in the brain structures of these rats [136,137]; however, comprehensive research on the biochemical and genetic features has not yet been conducted, and PM rats have not been tested in any preclinical studies.

**Table 3 biomedicines-11-01814-t003:** The strain-specific traits of PM rats.

Phenotype	Tests	References
Pendulum head and torso movements (100% of individuals)	Visual detection in a home cage	[84,135,137]
In up to 90% of individuals, audiogenic seizures (include wild running and/or generalized seizures)	Test for audiogenic epilepsy	[136,137]
Long postictal catalepsy	Test for audiogenic epilepsy	[137]
High excitability	Test for audiogenic epilepsy	[135,136,137,143]
Well-pronounced pinch-induced catalepsy in pups Increased vocalizations and motor paroxysms in pups	Test for pinch-induced catalepsy	[128]
Delay of the development of locomotor reactions and greater immobility in pups	Test for the activity of motor subsystems	[128]
Enhanced offensive behavior	Resident–intruder test	[91]
High aggressiveness	Glove test	[137]
Impaired spatial memory Lack of a spatial strategy in the Barnes maze Longer platform search time and a reduction in the proportion of successful attempts to find the platform in the Morris water maze	Barnes maze test Morris water maze test	[91,143]
Downregulation of norepinephrine and serotonin in the hypothalamus	Fluorometric quantitation of monoamines	[136]
Low concentration of taurine in the hippocampus	MRI (magnetic resonance imaging)	[102]

### 3.3. Epilepsy Modeling: Strategies and Approaches

Seizures and epilepsy types are usually subdivided into two categories: partial (focal) and generalized. Partial seizures can start from electrical activity in one area or group of cells on one side of the brain and may spread to other parts of the brain during the seizure, whereas generalized seizures are a result of excessive electrical discharges in both cerebral hemispheres at the same time [146]. A variety of animal species are used to study epilepsy, including fish, amphibians, and a wide range of mammals [131]. In the present review, special attention is given to rodent models of epilepsy.

The first animal models of epilepsy were generalized clonic convulsive seizures caused by direct electrical stimulation of the cerebral cortex in various mammalian species by David Ferrier in the late 19th century [147]. Other acute modalities that can trigger seizures include pentylenetetrazole injection [148,149]. Systemic administration of potent muscarinic agonist pilocarpine or kainic acid may lead to a prolonged period of spontaneous recurrent seizures [150,151]. All of the above models were set up by the induction of seizures in normal animals that were devoid of spontaneous seizures [152].

With advancements in gene-editing techniques, a variety of mice and rats have been identified as “epileptic” (experiencing spontaneous recurrent convulsions) or “seizure-susceptible” (having a low threshold for the acute initiation of convulsions) [131]. For example, “seizure-susceptible” dopamine D2 receptor knockout (D2R^−/−^) mice show increased susceptibility to kainic acid-induced seizures [153], as do Gpr39 (one of GPCR proteins) knockout mice [154,155] and Engrailed-2 knockout (En2^−/−^) mice, which display a gradual loss of dopaminergic neurons in the substantia nigra [156]. Genetic “epileptic” models include Lgi1 (leucine-rich glioma-inactivated 1) knockout rats, which are a model of autosomal dominant lateral temporal epilepsy [157,158]; Scn1a1^+/−^ (NaV1.1 sodium channel) knockout mice, which are a model of severe myoclonic epilepsy [159,160]; Sv2a (transmembrane glycoprotein) knockout mice, which experience lethal seizures [161]; 5-HT2C receptor–mutant mice, which also present infrequent and sporadic spontaneous seizures [162]; and other species (for a full review, see [163,164]).

The models described above are well-suited to investigation into the mechanisms and biomarkers of epileptogenesis or drug discovery that is targeted to certain genes; they may reveal treatments that are associated with already known etiopathogenesis pathways rather than uncover new ones [152].

Accordingly, genetic models that have arisen from the artificial selection of seizure-susceptible strains over many generations—resulting in high predisposition to epilepsy—are of particular interest. Such models include GAERS (genetic absence epileptic rats of Strasbourg) rats and WAG/Rij (Wistar albino glaxo rats from Rijswijk) rats. They emulate so-called human “absence epilepsy”, which involves brief generalized nonconvulsive seizures of sudden onset and abrupt termination [165,166,167,168].

Alt-hough the WAG/Rij and GAERS strains are better known as genetic models of absence seizures, they can serve as an audiogenic seizure model, a separate type of epilepsy induced by sensory stimulation (single acoustic stimulus). Other genetically selected reflex models that are susceptible to audiogenic seizures are GEPR, DBA/2, WAR, GASH:Sal, Krushinsky–Molodkina and, of course, PM rats, which are a special subject of this review. The genetically epilepsy-prone rat (GEPR) and dilute brown agouti coat color (DBA)/2 mice are models of reflex generalized tonic–clonic seizures [169]. The WAR strain is a genetic rat model; these rats are prone to audiogenic reflex epilepsy, acutely mimicking brainstem-dependent tonic–clonic seizures and chronically mimicking temporal lobe epilepsy [170]. GASH:Sal (genetic audiogenic seizure hamster from Salamanca) exhibits generalized tonic–clonic seizures that are characterized by a short latency period after auditory stimulation, followed by wild running, a convulsive phase, and finally stupor, with its origin being in the brainstem [171]. Krushinsky–Molodkina rats demonstrate a stable response to an audiogenic stimulus with a short latency period, which ends with a tonic–clonic seizure [172]. Such a large number of experimental options probably reflects the diversity of seizure types in humans [173].

All of the above models of audiogenic epilepsy involve so-called generalized (primary or secondary) clonic–tonic convulsions, which entail strong and sufficiently prolonged muscle rigidity (tonic convulsions) followed by rhythmic alternation of muscle contractions and relaxations (clonic convulsions). PM rats, in turn, experience abortive seizures, which resemble complex focal seizures with typical automatisms (aimless repetitive movements, such as stereotyped jumps reaching a height of 0.5 m at a speed of one jump per second) and do not result in generalized tonic–clonic seizures [137].

Locomotor agitation is the characteristic and most consistent component of audiogenic seizures in rodents, and it is a minimal convulsive response to a sound. Other components of audiogenic seizures (clonic and tonic seizures) may be absent, and PM rats represent this first component. More than 50% of PM rats experience these types of abortive seizures, ~20% experience a two-wave seizure with convulsions, and 10% manifest one running phase with tonic–clonic seizures [136].

## 4. Conclusions

In this review, our aim was to inform readers about unique animal models that have been designed in our laboratory through many years of breeding for various traits:

ISIAH (inherited stress-induced arterial hypertension) rats are a stress-sensitive model of arterial hypertension and are intended to help elucidate the genetic and physiological mechanisms of this disease. ISIAH rats can also be used to test and develop new antihypertensive drugs and new approaches for the treatment of arterial hypertension;

GC (genetic catatonia) rats exhibit catatonic reactions that are consistent with the key phenotypic traits of catatonic syndrome in humans. They can be utilized for researching the etiopathogenesis of catatonia, for identifying symptoms associated with catatonia and relevant psychiatric disorders, and for finding new molecular targets for the development of new drugs against catatonia;

PM (pendulum-like movements) rats present audiogenic abortive seizures, which resemble complex focal seizures with typical automatisms. A unique feature of PM rats is the presence of stereotypical (pendulum) movements of the head and shoulder girdle, which occur in response to even a weak stimulus. The catatonic signs of PM rats make it possible to study the comorbidity of symptoms of epilepsy and catatonic syndrome. They can be employed for investigations into the mechanisms of epileptogenesis, for studying the comorbidities of epilepsies (including catatonia, stereotypies, impulsiveness, and aggression), and for devising treatments that can reduce the propensity for audiogenic seizures.

Currently, transgenic models of diseases and pharmacological and surgical models are more popular due to their relatively quick and easy setup; however, breeding models have their undeniable advantages and are a very valuable tool for studying the genetic and physiological mechanisms of human pathologies.

## Figures and Tables

**Figure 1 biomedicines-11-01814-f001:**
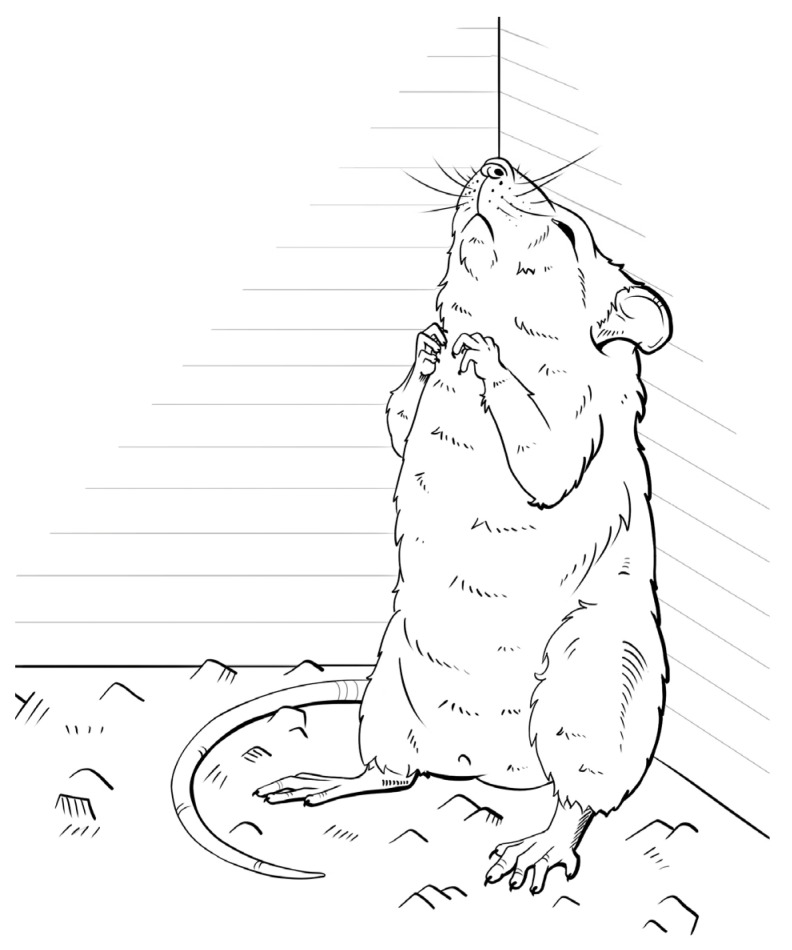
A GC rat in a cataleptic stupor.

**Figure 2 biomedicines-11-01814-f002:**
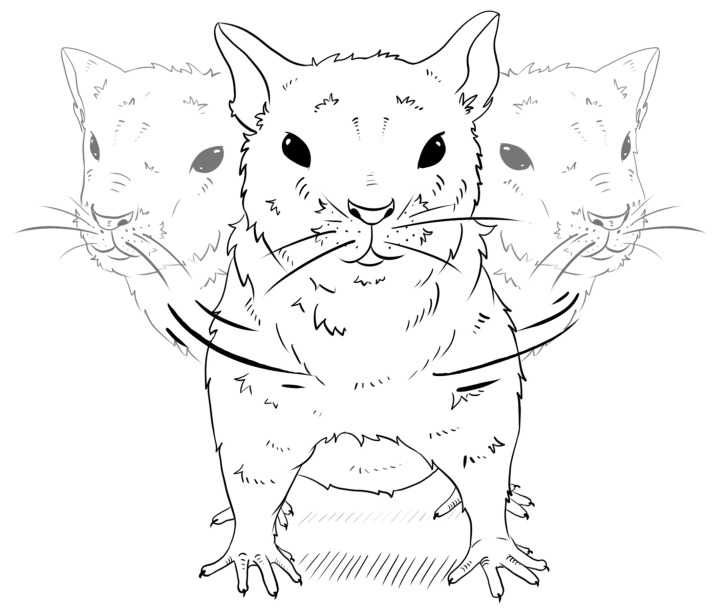
Pendulum-like movements of a PM rat.

**Table 1 biomedicines-11-01814-t001:** Strain-specific traits of ISIAH rats.

Phenotype	Approach	References
**Strain-specific traits**		
ISIAH rat strain breeding and general assessment [basal and stress-induced BP, hypertrophy of target organs (kidney and heart), age-dependent changes in basal and restraint stress–induced BP, age-dependent changes in activity of the hypothalamic–pituitary–adrenocortical system, age-dependent changes in dopamine and norepinephrine levels in brain structures (pons, medulla, hypothalamus, cortex)]	Genetic selection, BP measurement, body and target organs’ weight measurement, high-performance liquid chromatography (HPLC)	[2,3]
The structural organization of the adenohypophysis corresponds to an enhanced response of the hypothalamic–pituitary–adrenal axis in prehypertensive ISIAH rats	Electron microscopic analysis	[5]
Morphological signs of natriuretic peptide hypersecretion precede the development of genetically programmed high BP; in adult hypertensive rats, hypertrophic and degenerative changes in myocytes have been described	Electron microscopic analysis	[6]
Changes in hemodynamics and brain metabolites have been evaluated	Magnetic resonance imaging (MRI), MRI spectroscopy	[7]
Hypertrophy of renal corpuscles accompanied by structural changes that lead to an increase in the filtration barrier functional load and glomerular sclerosis	Electron microscopic analysis	[8]
Characteristics of the neurohormonal system	HPLC, immunohistochemistry	[4,9,10,11]
Behavior	The open field test and measuring the total activity in the home cage	[12,13,14]
Decreased bioavailability of nitric oxide in blood plasma	19F NMR measurement of NO production	[15]
Increased levels of triglycerides, very LDL and LDL cholesterols, a decreased content of HDL cholesterol, a high level of apolipoprotein B-100, and a decreased level of apolipoprotein A-I	Immunoblotting analysis	[16]
Increased basal activity of the central (brain) renin–angiotensin–aldosterone system (RAAS) in ISIAH rats. The RAAS is inhibited in the kidneys of adult ISIAH rats	Real-time PCR	[17,18,19]
Homozygosity of ISIAH rats	DNA fingerprinting	[20]
Genetic specificity of the ISIAH rat strain	Single nucleotide polymorphisms (SNPs)	[21,22]
**Steps toward drug discovery and translational medicine**		
A long-term reduction in basal and stress-induced BP has been obtained via injections of dopamine precursor L-DOPA during early development (21–25 days after birth).		[23]
The BP-lowering effect in ISIAH rats treated with reishi (*Ganoderma lucidum*) for 7 weeks is comparable with that of losartan. Unlike losartan, intragastric administration of reishi significantly increases cerebral blood flow.		[24]
Arginase inhibitor L-norvaline administered intraperitoneally (30 mg/kg) for 7 days to ISIAH rats causes a decrease in BP and an increase in diuresis.		[25,26]
A single intraperitoneal injection of nanocomposites containing antisense oligonucleotides (targeting *ACE1* or *AT1A* mRNA) conjugated with SiO_2_ or TiO_2_ nanoparticles leads to a decrease (pronounced within a week: ~30 mmHg) in systolic BP in ISIAH rats.		[27,28]
**Molecular markers of the hypertensive state in ISIAH rats**		
In two groups of male F_2_(ISIAH×WAG) hybrids at the ages of 3 and 6 months, genetic loci that are associated with traits related to the manifestation of the hypertensive status of ISIAH rats have been identified. The following has been analyzed: BP at rest and under short-term restraint stress; body weight; weights of target organs (kidneys, heart, and adrenal glands); plasma corticosterone concentration at rest and under stress; and behavior of the rats in the open field test. In a group of male F_2_(ISIAHxWAG) hybrids at an age of 6 months, QTLs for dopamine concentration in the brainstem, for norepinephrine concentration in the hypothalamus, as well as spleen weight were also determined.	Quantitative trait locus (QTL) analysis	[14,29,30,31,32,33,34,35]
A comparative analysis of the transcriptomes of the brainstem, hypothalamus, adrenal glands, renal cortex, and renal medulla has been carried out in hypertensive ISIAH rats and control (WAG) rats at the age of 3 months.	RNA-seq	[36,37,38,39,40,41]
Identification of candidate genes in genetic loci that are associated with BP and increased stress reactivity in ISIAH rats	QTL analysis RNA-seq	[42,43,44]
Identification of candidate genes that are associated with the manifestation of hypertensive status in ISIAH rats and changes in transcription levels during short-term restraint stress	RNA-seq Real-time PCR	[17,19,45,46,47,48,49,50]
Validation of candidate genes	Enzyme-linked immunosorbent assay (ELISA)	[51]

**Table 2 biomedicines-11-01814-t002:** The strain-specific traits of GC rats.

Phenotype	Tests	References
**Strain-specific traits**		
Cataleptic freezing (immobility and posturing/catalepsy) and hyperkinetic reactions (hyperexcitation: extreme nondirectional locomotor activity)	Test for catalepsy Open field test Light-dark box test	[85,86,90]
Impaired development of food-reinforced instrumental behavior	Instrumental conditioning	[90]
Altered social behavior in different situations Decreased social interactions in a new place Increased social exploration in a home cage	Three-chamber test Resident–intruder test	[91]
Slower solving of the Barnes maze	Barnes maze test	[91]
Increased startle reflex	SR-Pilot (San Diego Instruments) Startle response system (TSE)	[85,95]
Deficit of prepulse inhibition	Startle response system (TSE)	[96]
Altered maternal behavior	Visual registration in a home cage	[92]
High plasma corticosterone level Reduced plasma corticosterone level by chronic oral imipramine administration	ELISA kits	[98]
Smaller striatum area (in the right hemisphere) Larger cortex area (in the right hemisphere) Smaller area of anterior horns of lateral ventricles	MRI (magnetic resonance imaging)	[102]
**Steps toward drug discovery and translational medicine**		
Chronic per os administration of imipramine reduces cataleptic freezing	Test for catalepsy	[98,101]
Oral administration of the γ-polymorph of glycine reduces catalepsy, alleviates anxiety in the elevated plus maze test, and increases exploratory activity in rats in the light-dark box test Oral administration of both α- and γ-polymorphs of glycine enhances the exploratory activity of rats in the open field test	Test for catalepsy Open field test Light-dark box test Elevated plus maze test	[99]
Treatment with D-serine heightens anxiety and diminishes locomotor activity in the elevated plus maze test	Elevated plus maze test	[100]
**Molecular markers of catatonia**		
Decreased transcription of α1A adrenoreceptor in the medulla oblongata and midbrain Elevated transcription of α2A adrenoreceptor in the frontal cortex	Real-time PCR	[86,96]

## Data Availability

Not applicable.

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
