# Peer review of "Animal Models of Hypertension (ISIAH Rats), Catatonia (GC Rats), and Audiogenic Epilepsy (PM Rats) Developed by Breeding"

_biomedicines, 2023, doi:10.3390/biomedicines11071814_

Round 1

Reviewer 1 Report

Dear Marina A. Ryazanova and co-authors, 

Your review is based on the solid research and it is worth to publish. However, I suggest to focus on either cardiovascular-related topic with extensive data on  ISIAH strain or on catatonia- and pendulum movement strains related to mental disorders. It will help you to present data in more clear way, which would certainly benefit biomedical research. In any events, I also recommend to add such paragraphs in your revised review as "Drug Discovery" and "Translational Studies" using your animal models. 

 English needs to be improved to make the review more comprehensive 

Author Response

  • Your review is based on the solid research and it is worth to publish.

Answer: The authors thank the reviewer for the high evaluation of our work and comments, and suggestions that allowed us to improve the text of the manuscript. For the convenience of reviewers, all insertions in the text are written in red.

  • However, I suggest to focus on either cardiovascular-related topic with extensive data on ISIAH strain or on catatonia- and pendulum movement strains related to mental disorders. It will help you to present data in more clear way, which would certainly benefit biomedical research.

Answer: We discussed the reviewer's proposal to split the manuscript into two parts and present separately the hypertensive ISIAH strain or strains related to mental disorders. However, for a number of reasons, including financial ones, we would still like to present all three animal models in the current manuscript. For the convenience of readers, we have made a table of contents at the beginning of the text. In addition, we have inserted Tables that present the most relevant information for each model in a condensed form.

  • In any events, I also recommend to add such paragraphs in your revised review as "Drug Discovery" and "Translational Studies" using your animal models. 

Answer: Thanks for the suggestion.  The inserted Tables present not only the description of the strain specific traits, but also the section Steps to Drug Discovery and Translational Medicine.

  • Comments on the Quality of English Language:  English needs to be improved to make the review more comprehensive 

Answer: The text of the manuscript has been edited by a professional translator. The language certificate is enclosed.

Reviewer 2 Report

Dear authors,

The aim of this study was to review three rat models: arterial hypertension (ISIAH rats), catatonia (GC rats) and audiogenic epilepsy (MD rats), developed by breeding. The use of animal models of disease remains necessary for two primary reasons: a) to increase our knowledge about the disease, and b) to identify molecular targets that may lead to the synthesis of new drugs. Therefore, it is a topic that may interest many people.

However, locating the relevant information within this article can be challenging as it is presented solely within the text. Therefore, we suggest the introduction of a table for each model, containing the most relevant information (including bibliographical references), such as:

1) phenotypic characteristics of the model;

2) techniques used to demonstrate the validity of the model;

3) identified genes.

In addition to the tables, we would also like to know, for each model (some have it, others do not):

1) when the selection of each model started and when it was considered finished;

2) after how many weeks/months the model is ready to be used (depending on how soon it presents phenotypic manifestations, or before that?);

3) are there any differences between males and females?

Thank you for the work you presented. Good luck and success.

Regarding the text, in line 291, do you mean "benzodiachepines" or "benzodiazepines"?

Author Response

The aim of this study was to review three rat models: arterial hypertension (ISIAH rats), catatonia (GC rats) and audiogenic epilepsy (PM rats), developed by breeding. The use of animal models of disease remains necessary for two primary reasons: a) to increase our knowledge about the disease, and b) to identify molecular targets that may lead to the synthesis of new drugs. Therefore, it is a topic that may interest many people.

The authors thank the reviewer for the high evaluation of our work and comments, and suggestions that allowed us to improve the text of the manuscript. For the convenience of reviewers, all insertions in the text are written in red.

1) However, locating the relevant information within this article can be challenging as it is presented solely within the text. Therefore, we suggest the introduction of a table for each model, containing the most relevant information (including bibliographical references), such as:

1) phenotypic characteristics of the model;

2) techniques used to demonstrate the validity of the model;

3) identified genes.

Answer:  As suggested by the reviewer, we have inserted information tables that present the most up-to-date information for each model in a concise manner. These tables provide a description of strain-specific traits, as well as a section on "Steps towards drug discovery and translational medicine", as well as a section presenting the main results of molecular genetic research aimed at searching for molecular markers.

2) In addition to the tables, we would also like to know, for each model (some have it, others do not):

1) when the selection of each model started and when it was considered finished;

2) after how many weeks/months the model is ready to be used (depending on how soon it presents phenotypic manifestations, or before that?);

3) are there any differences between males and females?

Answer: The available information was included in the text of the manuscript and Tables.

Thank you for the work you presented. Good luck and success.

Comments on the Quality of English Language:

Regarding the text, in line 291, do you mean "benzodiachepines" or "benzodiazepines"?

Answer: The text of the manuscript has been edited by a professional translator. The language certificate is enclosed.

Round 2

Reviewer 2 Report

Dear authors,

The changes made to the manuscript have made it clearer and easier to navigate.

Thank you for the work you presented.

Good luck and success.